



# Origin, transport and processing of organic aerosols at different altitudes in coastal Mediterranean urban areas

Clara Jaén[1,2], Mireia Udina[3], Roy Harrison[4], Joan O. Grimalt[1], and Barend L. Van Drooge[1]

[1] Institute of Environmental Assessment and Water Research (IDAEA-CSIC), Jordi-Girona, 18-26, Barcelona, 08034, Catalonia, Spain

[2] Doctoral Programme in Analytical Chemistry and the Environment, University of Barcelona, Faculty of Chemistry, Martí i Franquès 1-11, 08028, Barcelona, Spain

[3] Department of Applied Physics—Meteorology, University of Barcelona, C/ Martí i, Franquès, 1, 08028 Barcelona, Catalonia, Spain Doctoral

[4] Division of Environmental Health and Risk Management, School of Geography, Earth and Environmental Sciences University of Birmingham, Edgbaston, Birmingham, B15 2TT, United Kingdom

*Correspondence to*: Barend L. Van Drooge (barend.vandrooge@idaea.csic.es)

**Abstract.** Organic molecular markers in atmospheric PM10 were analysed by off-line GC-MS techniques in an urban background site (81 m above sea level (asl)) and in a nearby elevated sub-urban background site (415 m asl), in cold and warm periods in Barcelona; situated in the Western Mediterranean Basin. Previous studies reported similar PM concentrations and substantial organic matter contributions in both sites but did not analyze the organic molecular composition, which is expected to vary within the city's vertical airshed due to a weakening influence of local emission sources and enhanced influence of regional air masses. Multi-variant analysis of organic molecular marker concentrations, together with major air quality parameters (NO, $NO_2$, $O_3$, $PM_{10}$), resolved six components that represented primary emissions sources and secondary organic aerosol formation processes: 1) diurnal traffic 2) nocturnal traffic, 3) biomass burning, 4) biogenic with primary and secondary organic markers, 5) fresh secondary, and 6) regional secondary. Urban traffic emissions reached the elevated site during daytime through the sea-mountain breeze, while nocturnal traffic emissions accumulated in the nighttime urban atmosphere, when the two sites were often disconnected by temperature inversions. Biomass burning, dominant in the cold period, was the main contributor to toxic PAHs in these two background sites. Regional secondary organic aerosol contribution was more abundant in the elevated background site. Several SOA formation mechanisms were identified such as the oxidation of traffic emissions by $NO_x$, the aqueous-phase oxidation under high relative humidity, and formation of fresh SOA under conditions of low relative humidity.





## 1 Introduction

Particulate matter (PM) interacts with solar radiation modifying the Earth's radiation balance and involving deleterious effects on the ecosystems and human health (Garland et al., 2008; Malavelle et al., 2019; Soleimanian et al., 2020). Chronic PM exposure has been related with increased incidences of stroke, lung cancer, asthma, type II diabetes, myocardial infarction, and mental illnesses (Chen & Hoek, 2020; IARC, 2013; Janssen et al., 2013; Lida et al., 2017; Manisalidis et al., 2020; Tsai et al., 2019).

Airborne PM is generated by primary sources such as traffic, industries, wildfires, and formed secondarily in the atmosphere by photochemical transformations, nucleation or condensation processes of smaller particles or gas-phase compounds in the atmosphere. Both primary and secondary PM can have natural or anthropogenic origins, and their relative contributions depend on the characteristics and intensity of local emissions as well as on the chemical and physical atmospheric conditions. The most important PM constituents are inorganic salts, crustal and anthropogenic minerals, elemental carbon, and organics (Brines et al., 2019; Harrison, 2020; Myhre et al., 2009; Tao et al., 2021).

The contribution of organic aerosol (OA) to PM varies from 20 % to 80 % (Jimenez et al., 2009; Murphy et al., 2006; Putaud et al., 2004; Zhang et al., 2007). Primary organic aerosols (POA) include marker compounds such as polycyclic aromatic hydrocarbons (PAHs) generated during incomplete combustion of biomass or fossil fuels, hopanes from mineral oils, or anhydrosaccharides from biomass combustion. Conversely, the oxidation of volatile organic compounds (VOC) leads to the formation of the secondary organic aerosols (SOA) (Atkinson, 2000; Odum et al., 1996; Palm et al., 2018; Srivastava et al., 2022). This oxidation implies an addition of oxygen and/or nitrogen atoms to the molecular structures reducing the volatility. The VOCs undergoing these processes can be either from biogenic origin such as isoprene and α-pinene or derived from anthropogenic activities like volatile aliphatic and aromatic hydrocarbons, which can be very relevant in Mediterranean cities (Minguillón et al., 2016; van Drooge et al., 2018).

The presence of multiple PM emission sources, combined with unfavorable meteorological conditions, often leads to poor air quality in urban environments. The most severe air quality events typically occur during strong anticyclonic conditions, which promote temperature inversions that trap surface-emitted primary pollutants within shallow mixing layers. Under these conditions, ozone ($O_3$) can accumulate in nocturnal residual layers above the inversion (Jaén, Udina, et al., 2021; Massagué et al., 2021). The accumulation of this strong oxidant may also enhance SOA formation in these upper layers as well as during daytime due to increased photochemical activity. This phenomenon is especially relevant in the Western Mediterranean Basin (WMB) that has particular air pollution dynamics due to its location between mid-latitudes and subtropical regimes and the influence of the Mediterranean Sea (Derstroff et al., 2017).

Over the last decade, several studies carried out in the metropolitan area of Barcelona gave insight on the spatial and vertical evolution of particles and other air quality parameters in the urban airshed, although the molecular organic composition was only described at the lower city level (Alier et al., 2013; Brines et al., 2019; Dall'Osto et al., 2013; van Drooge et al. 2018). These studies showed that traffic related pollutants were mainly attributed to local emission sources, while pollutants from



biomass burning, harbour and industrial emissions, as well as compounds related to secondary aerosol formation processes, were of regional origin. Concerning vertical mixing within the urban airshed, the previous studies suggested that the air column above the city presents conditions that promote new particle formation (NPF) events and SOA formation due to a presumed weakening influence of local source emissions to PM. Although the higher $PM_1/PM_{10}$ ratios that were observed in the elevated sites above the city can be related to these aerosol formation processes (Dall'Osto et al., 2013), it was not clear to what extend local emissions other than traffic, such as biomass burning emissions, and recirculation of regional air masses influence these elevated urban background sites.

Organic matter is an important contributor to $PM_{10}$ in the urban background sites, but detailed information on the molecular organic compound composition at various altitudes is limited and has not been studied yet in this vertical urban Mediterranean setting. The present study aims to characterize the POA emissions and accumulation conditions, as well as SOA formation processes in the urban background atmosphere of Barcelona through the analysis of molecular organic makers in $PM_{10}$ in two altitudes (81 m above sea-level (m asl) and 415 m asl at 12-hour intervals, covering day and nighttime over one week in warm and cold periods.

## 2 Materials and methods

### 2.1 Sampling site

Barcelona is located on the north-eastern coast of the Iberian Peninsula in the Western Mediterranean Basin. This region is a model case of environmental challenges in transition zones between temperate and arid climates. The WMB experiences intense solar radiation throughout the year, leading to high photochemical activity, even in winter. This solar activity interacts with the biogenic VOCs emitted by the region's abundant vegetation, resulting in the formation of large amounts of SOA (Ciccioli et al., 2023; van Drooge et al. 2018). Additionally, the PM mass and composition in the WMB are often altered by the transport of mineral dust from the arid regions of North Africa worsening the air quality in the basin (Querol et al., 2009; Schepanski et al., 2016).

The synoptic and regional meteorological scale affecting the WMB is mainly driven by the Azores high-pressure system, the Saharan and Iberian low-pressures, the topography, and the heat-moisture regulation by the Mediterranean Sea. The interactions between these factors lead to highly contrasting temperature, humidity and rainfall conditions among seasons. Barcelona is geographically limited by two rivers on the east and west sides, by the Mediterranean Sea in the south and by the Collserola hills in the north (512 m asl). Due to this orography, the city is typically influenced by sea breezes that flow inland during the day as solar heating enhances the vertical ascent of warmer air over the city creating a surface horizonal thermal gradient. At night, the opposite occurs as the cooler air above the city subsides. In the absence of large-scale meteorological phenomena, this can create a recirculation pattern, leading to distinct air masses at the surface and upper levels.

The city of Barcelona is home to 1.7 million inhabitants, while its metropolitan area hosts around 3.7 million people, involving one of the highest population densities in Europe (16.339 hab./km$^2$). The car density in the city is around 6.000/km$^2$ with a



daily vehicle flux of 350.000 in the city centre and up to 151.000 in the main ring roads (Barcelona city council., 2020). Moreover, the activities in the nearby harbour and airport, the agricultural works in the Llobregat river delta and the emissions from several industrial zones and power plants in the metropolitan area are additional potential sources influencing the air quality of the city (Dall'Osto et al., 2013).

In order to assess how the interactions between these specific meteorological conditions and multisource particle emitters

determine the organic composition of airborne aerosol in the city, $PM_{10}$ was collected at two background sampling stations were installed at two altitudes (Figure 1). One, at 81 m asl, in the urban background site of IDAEA (city site), and the other, at 415 m asl, situated on a hill top of the Collserola hills overlooking the city (elevated site). The sites are separated by a horizontal distance of 3.5 km. The urban background site is situated within the city airshed while the atmosphere in the elevated site can be located either within the urban airshed or above the atmospheric layer influenced by the city in the case of low

planetary boundary layer height (PBLH) because of strong temperature inversions.

**2.2 Sampling methodology**

$PM_{10}$ samples were collected at 12 hours intervals on 150 mm-diameter quartz filters (Pall Corporation, USA) by means of a high-volume sampler (DHA-80, Digitel, Switzerland) equipped with a $PM_{10}$ inlet. Filters were pre-baked at 450 ℃ before use. The instrument air flow was 500L/min, providing a final sample volume of 360 $m^3$. Sample collection was performed

simultaneously at both sites during two one-week campaigns in the warm and cold periods (April 26-May 3, 2022 and January 31-February 6, 2023, respectively). Daytime samples were taken from 7:00 to 19:00 UTC (9:00 to 21:00 and 8:00-20:00 local time during the warm and cold periods, respectively). The nighttime samples were collected from 19:00 to 7:00 UTC.

**2.3 Chemical analysis**

The analytical procedure to determine the concentration of organic compounds in the samples was performed as described in

previous studies (Fontal et al., 2015; Jaén et al., 2023; Jaén, Villasclaras, et al., 2021). Briefly, two 45 mm-diameter punches of the whole filter were spiked with deuterated PAH (phenanthrene-d10, anthracene-d10, fluoranthene-d10, pyrene-d10, benzo[a]anthracene-d12, chrysene-d12, benzo[b]fluoroanthene-d12, benzo[k]fluoroanthene-d12, benzo[a]pyrene-d12, indeno[1,2,3-cd]pyrene-d12, dibenzo(a,h)anthracene-d14, benzo[ghi]perylene-d12), succinic acid-d4 and levoglucosan-d7. Then, the filter fractions were extracted with a dichloromethane: methanol 1:1 v/v mixture by ultra-sonication ($3 \times 10$ mL; 15

min). The obtained extracts were filtered with glass-microfibre discs and concentrated to 0.5 mL by means of roto-evaporators and with a nitrogen stream. A 25 µL aliquot of the extract was transferred to a conical vial, evaporated to dryness and treated with 25 µL of BSTFA and 10 µL of pyridine to obtain the trimethylsilyl derivatives of the compounds with hydroxyl groups. This aliquot was injected to a GC-MS (8290 GC with 5975 MSD, Agilent Technologies) in full scan mode to quantify the polar compounds (isoprene or α-pinene SOA products, dicarboxylic acids and saccharides). A liquid-liquid extraction was

performed with the remaining extract with n-hexane ($3 \times 0.5$ mL) to analyse the non-polar fraction (polycyclic aromatic hydrocarbons (PAHs), their methyl (mPAH) and oxygenated (oxyPAH) derivatives, alkanes and hopanes). The hexane extract



was concentrated with a nitrogen stream to a final volume of 25 µL prior to its injection in a Q-Exactive GC Orbitrap MS (Agilent Technologies) in full scan mode. The chromatographic separation was obtained with a HP-5MS 60 m capillary column for the polar compounds and with a HP-5MS 30 m column for the non-polar.

Two additional 45 mm-diameter punches of the filter were spiked with 9-nitroanthracene-d9 and extracted with 7 mL of DCM by ultra-sonication during 20 min for the analyses of nitro derivatives of PAH (nitro-PAH). The extract was concentrated with a nitrogen stream and then filtered with anhydrous sodium sulphate contained in a Pasteur pipette with glass wool. The filtered extract was further evaporated to 50 µL with a nitrogen stream, 50 µL of nonane were added to the extract and the concentration continued to evaporate the remaining DCM. The extracts were analyzed by GC-MS using an Agilent 5973N Inert EI/CI Mass

Spec Selective Detector with a Agilent 6890N Gas Chromatograph) operated in SIM and in negative ion chemical ionisation (NICI) mode equipped with a Restek Rxi-5Sil MS 60 m column (2022 campaing) or by GC–MS/MS using an Agilent 7000 Series Triple Quad) equipped with a HP-5MS 60 m capillary column (2023 campaign). In this last case, the instrument operated under electron impact ionization and multiple reaction monitoring (MRM) mode was used for acquisition. These instrumental methodologies provided equivalent nitro-PAH results.

Sixty-eight compounds were identified and quantified based on their chromatographic retention times, mass spectra and calibration with analytical internal standards. Field blanks of each sampling campaign and site were collected and treated together with the sample filters. The average blank levels were subtracted from the sample concentrations. Methods limit of detection (MLOD) were determined as the average blank level plus three times the standard deviation of blanks or by the instrumental limit of detection (ILOD) in case of blanks below this limit. ILOD was set based on a minimum signal-to-noise

ratio of 3. To process the data, concentrations below MLOD were replaced by half of this value.

**2.4 Air quality and meteorological data**

NO, NO$_2$, O$_3$ and PM$_{10}$ hourly values were obtained from the two air quality stations of the Atmospheric Pollution Monitoring and Forecasting Network situated in the same sites as the PM$_{10}$ samplers: *Palau Reial, IDAEA-CSIC,* for the city site and *Observatori Fabra* for the elevated sub-urban site (Generalitat de Catalunya, 2020) (Figure S1).

Meteorological data, temperature, pressure, relative humidity, precipitation, solar radiation, mean wind velocity (10 m), mean wind direction (10 m) and solar radiation, were obtained every half an hour from two Automatic Weather Stations of the Catalan Meteorological Service (Generalitat de Catalunya, 2019). This information was measured in the elevated and low-level sampling sites. The former was located besides the elevated sampling location and the latter at 1.2 km distance of the city site (Figure S1). Complementary meteorological data nearby the Barcelona harbour was obtained from a station located

in this installation (Figure S1 and Figure S2).

PBLHs were estimated from a CL31Vaisala Ceilometer set located at 98 m asl and at 340 m in horizontal distance from the city site (Figure S1). The original data was processed using the Vaisala Boundary-Layer View software (BL-VIEW) Enhanced Gradient method (García-Dalmau et al., 2021; VAISALA, 2020) followed by a selection algorithm based on the methodology of Lotteraner and Piringer (2016) to obtain PBLH values every 10 minutes.



### 2.5 Air mass trajectories and clustering

Backward air trajectories for 96 h were computed with the NOAA HYSPLIT model (Rolph et al., 2017; Stein et al., 2015) provided with NCEP/NCAR Global Reanalysis Data (Kalnay et al., 1996) with a horizontal resolution of 2.5 degrees as meteorological files. The trajectories were computed hourly for the two sampling campaigns at the city site at 100 m above ground level (agl), at the elevated site at 400 m agl, and at the middle point between both sites at 200 m agl as starting locations. Moreover, a 48 h-cluster analysis joining the two sets of trajectories (the two sampling periods) was performed to elucidate the air mass sources reaching Barcelona at a mesoscale to synoptic meteorological scale. This analysis provided the origins of air masses during the sampling periods. The clustering algorithm groups similar trajectories in clusters and represent them by their mean trajectory. The optimal number of clusters was 5 considering the percentage change in total spatial variance through a step-wise reduction in cluster numbers (92 % from 5 to 4 clusters in the last case). Moreover, the geographical interpretation of the clusters was also taken into consideration. Given the proximity between sampling sites, a lack of significant differences between different altitudes of ending point back-trajectories was observed (Figure S3). Therefore, the middle point (Figure S3 b) was used for the work description.

## 3 Results and discussion

### 3.1 Automatic data and air mass origin

The temporal evolution of potential temperature (θ) during both sampling campaigns exhibited similar trends at both altitudes. During the day, temperature and relative humidity (RH) remained comparable at the two sites, indicating that they were within the convective mixing layer of the urban airshed. However, temperature peaks at the elevated site appeared slightly delayed compared to the city site due to the progressive increasing of the mixing layer in the hours following sunrise. At night, more pronounced differences in temperature and RH were observed between the two sites, indicating a disconnection of the air mass at the elevated site from the one in the urban airshed. Generally, θ was higher at the elevated site showing the stability of the atmosphere under nighttime temperature inversions. This day-night dynamic was also observed in the differences between wind speed and wind directions; being similar during the day, but stronger and from a different direction in the elevated site compared to the city site during nighttime (Figure S4).

Regarding the air quality parameters, this nocturnal decoupled behaviour was also observed for ozone that showed a very weak correlation between sites at night, whereas a good correlation was found during the day ($r^2$=0.7) (Figures S4 & S5). This oxidant remained high at the elevated background site during all day (78±15 µg/m$^3$), while in the city site, O$_3$ exhibited a diurnal oscillation peaking during the day (47±19 µg/m$^3$). This agrees with previous observations in the same sampling sites at different seasons (Dall'Osto et al., 2013) and suggest that the different oxidant distribution in the vertical column may alter the organic composition at the studied altitudes. The nitrogen oxides also showed substantial correlations between sites during the day, although always higher in the city, but they were uncorrelated at night (Figures S4 & S5). Their diurnal trends at the





city site were strongly linked to traffic emissions, with pronounced peaks in the early morning and late afternoon (up to 107 µg/m$^3$) but those high concentrations reached the elevated site only under mixing conditions of the atmosphere (25±19 µg/m$^3$ city site; 10±10 µg/m$^3$ elevated site). PM$_{10}$ had moderate correlations between sites during both periods of the day (r$^2$=0.32-035, Figures S4 & S5) and the overall concentration trends remained similar (18±7 µg/m$^3$ city site; 16±7 µg/m$^3$ elevated site), although the slightly higher warm-period PM$_{10}$ at the elevated site may be another indication for variations in their organic composition.

The 5 clusters of 48h-backward trajectories for both sampling periods resolved with the HYSPLIT clustering algorithm are showed in Figure S3 b. The first (warm) sampling period was dominated by southerly trajectories (red cluster) and trajectories coming from the east (dark blue cluster). Both clusters have their origins in the Mediterranean Sea indicting regional recirculation. The end of this period was dominated by northerly trajectories coming from the European continent (light blue cluster). This change in the air mass origin from regional to continental was reflected in the air quality (Figure S4) involving a reduction of NO, NO$_2$ and PM$_{10}$ peaks when the air was from the north. In these conditions, the differences in O$_3$ between both sampling sites were reduced. In the second (cold) sampling campaign, all trajectories reached Barcelona from the north, the first period coming more directly from the Atlantic Ocean (green cluster) while trajectories coming from the gulf of Lion were observed in the last sampling days (pink cluster). This change was again reflected in lower pollutant concentrations at the end of the campaign.

**3.2 Organic aerosol composition**

Table 1 shows detection frequencies (DF), MLOD, average, maximum and minimum concentrations at the city and elevated background sites and p-values for the Wilcoxon test for paired samples (WPS) between both sites, and average day and nighttime concentrations are given in Table S 1. The magnitude of these differences between sites are evaluated by calculating the Decreasing Index for each compound in each pair of samples as following: $DI_i = \frac{C_{elevated\ site\ i} - C_{urban\ site\ i}}{\sum_1^n C_{j}/n}$; where $i$ is the specific pair of samples, and n the total number of samples. The average DIs for day and night samples are shown in Figure 2. The average concentrations of individual PAHs ranged from 8 to 230 pg/m$^3$ at the city site and from 6 to 152 pg/m$^3$ at the elevated site. These compounds are generated by incomplete combustion, mainly from biomass and fossil fuels. Their concentrations were higher in the city site compared to the elevated site (p<0.05). Similar trends were observed for the methyl-PAHs which are usually emitted alongside PAHs under relatively low temperature combustion conditions. The average levels of individual methyl-PAHs varied between 5 and 183 pg/m$^3$ at the city site and between 2 and 157 pg/m$^3$ at the elevated site. Retene (RET), which is an indicator of pinewood combustion (Ramdahl, 1983), was the more abundant methyl-PAH, with similar concentrations as predominant unsubstituted PAHs (pyrene (PYR), benzofluoranthenes (BBJKFL) and benzo[ghi]perylene (BGHIP)), which suggests contribution of pinewood combustion in the sites (Van Drooge & Grimalt, 2015).



As regards to oxygenated PAHs, the average concentrations of the individual compounds ranged from 19 to 314 pg/m$^3$ and from 10 to 289 pg/m$^3$ at the city and elevated sites, respectively. The most abundant compound was 9,10-anthraquinone (ANQ) followed by 2-methylanthraquinone (2mANQ) and fluoren-9-one (9FLO), which have been associated with several primary

combustion emissions, but also to secondary transformation processes, such as the reaction of anthracene and fluorene with OH, NO$_x$ or O$_3$ (Keyte et al., 2013; Kojima et al., 2010; Oda et al., 1998; Rogge et al., 1993). This dual origin probably explains their similar concentration at both sampling sites. The other oxyPAH are significantly higher in concentration at the city level. The two nitro derivatives of PAHs reported in this study exhibited relatively low concentrations, but with higher 9-nitroanthracene concentration in the city site than the elevated site (11 and 5 pg/m$^3$, respectively). These compounds can be

primary emitted from combustion sources, but also formed from anthracene reactions in the presence of NOx (Bandowe et al., 2014; Saldarriaga et al., 2008), and their significant higher concentrations in the city site indicates a priory a dominant influence of primary emissions over formation processing.

In almost all cases, the decreasing indexes of PAHs and derivatives were negatives for both day and night periods agreeing with higher concentrations at the city level (Figure 2). However, the DIs were more accentuated for nighttime samples, up to

-1.3, indicating larger differences between sites at night when air masses are disconnected.

The hopanes are traffic markers for their presence in lubricant oils (Schauer et al., 2007), showed an average concentration of 117 pg/m$^3$ at the city site and 42 pg/m$^3$ at the elevated site (Table 1), with negative DIs for both day and night samples (Figure 2).

The anhydro-saccharides galactosan (GAL), mannosan (MANNO) and levoglucosan (LEV) are marker compounds of biomass

burning (Simoneit, 2002). They showed predominance of LEV, with average concentrations at the city and elevated sites of 80 and 66 ng/m$^3$, respectively. The concentrations in the cold period (up to 300 ng/m$^3$) were ten times higher than the warm period, which is similar to results in previous studies in the urban area of Barcelona (Reche et al., 2012; van Drooge et al., 2018). Nighttime DIs were around -0.4, indicating an accumulation of these combustion products in the city while they were close to 0 or positive during the day, pointing to vertical mixing in the urban airshed (Figure 2). The substantial concentrations

of these biomass combustion markers at the elevated site at nighttime in the cold period suggest an influence of local emissions besides and influence of recirculation of biomass burning aerosols (BBOA) in the region.

The 12 dicarboxylic acids (DCA) that were analysed in this study show different trends than the combustion products mentioned before. The C$_2$-C$_9$ linear homologues, two hydroxylated compounds (malic acid (MA) and tartaric acid (TARA)) and phthalic acid (PHA) are mainly formed through photochemical reactions of VOCs from natural and anthropogenic primary

emission sources such as vehicular exhaust, food cooking, or biogenic emissions (Cao et al., 2017; Kunwar & Kawamura, 2014; Lui et al., 2023; Rogge et al., 1991; Xu et al., 2020). Their average individual concentrations span from 0.3 to 17 ng/m$^3$ at the city site and from 0.3 to 21 ng/m$^3$ at the elevated site. Malonic acid and tartaric acid were higher at the elevated site, while pimelic, suberic, azelaic and phthalic acids were more abundant at the city site. In previous studies, these later longer chained C$_7$-C$_9$ DCAs had higher concentrations in an intensive traffic site in the populated city centre and were linked

to the oxidation of unsaturated carboxylic acids, such as oleic acid (Alier et al., 2013, Kawamura and Gagosian, 1987). Other





dicarboxylic acids, such as succinic acid, glutaric acid, and phthalic acid did not show statistically significant differences in concentration between the two sites.

Secondary organic aerosol markers from biogenic volatile compounds, i.e. isoprene (methyltetrols (MT) and 2-methyl glyceric acid (MGA)) and α-pinene (cis-pinonic acid (CPA), 3-methyl butane tricarboxylic acid (MBTCA) and 3-hydroxyglutaic acid

(HGA)) oxidation products (Claeys et al., 2004, 2007; Szmigielski et al., 2007), ranged from 1 to 14 ng/m$^3$ at the city site and from 1 to 27 ng/m$^3$ at the elevated site. Their concentrations were around ten times higher in the warm period compared to the cold period, relating these compounds to the period with highest biogenic (vegetation) activity and photo-chemical reactions. Higher levels were observed at the elevated site compared to the city site, with the largest difference for CPA which was also the most abundant one. CPA is a fresh SOA from α-pinene oxidation product while 3HGA and MBTCA are further oxidation

products (Claeys et al., 2007; Szmigielski et al., 2007). CPA decreasing index was 0.4 for daytime samples and 0.8 for nighttime samples, suggesting further oxidation of CPA in the city compared to the elevated site. This may be related to enhanced formation of aged pinene SOA in the presence of $NO_x$ from traffic emissions as has been observed in previous studies in the urban background in Barcelona (Alier et al., 2013; Minguillón et al., 2016; van Drooge et al., 2018). On the other hand, the presence of pine forests in the Collserola hills may also be a potential source for α-pinene emissions and CPA formation.

Lastly, other biogenic saccharides were also determined in this study which can be related to fungal sporulation (mannitol (MANNI) and sorbitol (SOR)), pollen grain and organic soil dust (glucoses (AGLU and BGLU) and meso-erythritol (MERY)) which are relevant in the PM coarse fraction (Bauer et al., 2008; Burshtein et al., 2011; Jia & Fraser, 2011; Medeiros & Simoneit, 2008; Simoneit et al., 2004). Their individual average levels ranged between 1 and 24 ng/m$^3$ in the city site and between 1 and 42 ng/m$^3$ at the elevated site. Generally, they showed higher concentrations at the elevated site agreeing

with the higher density of vegetation in this site compared to the city site. However, this difference was only significant for MANNI, AGLU and BGLU which showed high positive DIs during the day (0.8-1; Figure 2).

Table S2 compares the concentrations of the studied compounds at both sites and campaigns with similar studies in the region in urban, sub-urban, rural, industrial, and high-altitude areas. In general, the levels of the studied compounds were in the range of these former studies in terms of location and season. In particular, the concentration of combustion markers (biomass

burning markers, PAHs and derivatives) in the cold period campaign were similar to urban and sub-urban sites in cold periods, but lower than those registered in rural areas with local biomass burning emission sources. The hopanes were lower than recorded in suburban, industrial and traffic sites under stagnant atmospheric conditions, while the biogenic SOA marker concentrations were lower than those observed in rural sites.

With respect to nitro-PAHs, few studies report their PM concentrations in southern Europe. In fact, the only measurements of

these compounds in ambient air in the city of Barcelona are those from Bayona et al., 1994 in heavily trafficked sites of the city. They reported much higher concentrations of 9-nitroanthracene and 2-nitrofluorene than detected in the present study, which is also reflected in concentrations of the parent PAHs. In contrast, the oxyPAH ANQ, which was also reported in both studies, was found in lower concentration in the early 90's. More recent studies in European cities also found higher concentrations of nitro-PAHs than those observed in the present study, particularly in winter (Alam et al., 2015; Alves et al.,



2017; Tomaz et al., 2016). However, the concentrations observed in an urban area on the Iberian Peninsula were more similar (Lara et al., 2022).

## 3.3 Apportionment of sources and atmospheric processes

The multi-variant analysis by MCR-ALS was applied on the joined dataset, and included the organic markers in $PM_{10}$ samples, as well as the average concentrations of the air quality parameters NO, $NO_2$, $O_3$, and $PM_{10}$ for the sample periods. This was
done in order to study the similarities and differences among the quantified compounds and sampled sites and to identify the most significant sources and atmospheric processes influencing air quality in the sites without pretending to be a mass balance for PM, or the organic aerosol. The analysis resolved six components based on their chemical composition in relation to their environmental interpretability. Those components explained 95 % of the variance in the dataset. The components are represented in Figure 3 showing the loadings of each compound in each component (bars) and the percentage of the compound
present in the specific component as the proportion of the compound along all components (purple dots). Moreover, Figure 4 shows the score of the resolved components in each sample. The six components are as follows:

### 3.3.1 Diurnal traffic

The first component (18 % of explained variance) accounted for a high percentage of the loadings of traffic markers, such as NO (62 % of total NO loadings), hopanes (50 %), and $NO_2$ (36 %), but moderated contribution of PAHs (10 %) and mPAH
(20 %; excluding RET). Longer-chained dicarboxylic acids such as PHA, AZA, SUBA, PIMA and ADIA, as well as oxygenated PAHs 2mANQ, 23dmANQ and BAF, and 2-nitrofluorenone (46 %), were also represented in this component (Figure 3). These compounds can originate directly from traffic emission sources but are also related to fast oxidation of traffic related VOCs and those from food cooking during atmospheric transport (Alier et al., 2013). The highest scores of this component were found in the city site during daytime, which is consistent with the urban inputs and photo-chemical reactions,
and SOA formation. This component probably resolves part of the fresh urban inputs (traffic and cooking), since an important time frame of the traffic rush hours and cooking period peak are during the nighttime period (Figure S4). Nevertheless, other sources, such as those from the harbour area could also be involved, since they are usually characterized by elevated concentrations of $NO_2$, but also $SO_2$. Barcelona harbour is situated in the southern coast (Figure S1) and winds from the sea can transport these emissions towards the city site, which is probably the case in the first days of the warm period campaign.
Figure S2 of the supplementary material shows that during the day wind comes predominantly from the south, indicating an inland transport of the visible pollution from the harbour (photos in Figure S6). This is also supported by the $SO_2$ peaks observed at the city site (Figure S2) during daytime sampling in both sampling campaigns.





### 3.3.2 Nocturnal traffic

The second component (13 % of the explained variance) had the highest loadings for the methyl phenanthrenes (51 % of the total loadings of methylphenenthrenes) and 2mANT (66 %). The occurrence of these compounds in urban areas is usually related with fossil fuel emissions, particularly from diesel engines (Casal et al., 2014), pointing to a traffic emission related component. It had higher scores at nighttime (Figure 4) which may partially be the morning rush hour traffic, which explains the high loadings of PAHs (25 %), hopanes (19 %), $NO_2$ (32 %), NO (21 %) and methylchrysenes (31 %) (Figure 3). Moreover,

PAH derivatives, ANQ, 2mANQ and 9nANT also have high loadings in this component in agreement with previous studies that linked them to traffic sources (Oda et al., 1998; Saldarriaga et al., 2008).

Consistent with traffic emission origin, this component was higher in the city site than in the elevated site. Besides the early morning rush-hour contribution, this component may reflect various activities that involve traffic emissions, such garbage collection, street cleaning and transport of goods during the nighttime with trucks. The high NO and $NO_2$ loadings in this

component further support this association. It is worth noting that the Low Emission Zone regulations in Barcelona are valid on working days from 7 am to 8 pm, allowing the more polluting vehicles (old diesel vehicles) to circulate at night in the city. Moreover, this component had the highest scores on the four first nights when the air masses were indicating more stagnant conditions. The lowest scores were observed during weekends, when vehicle movements are reduced, reinforcing the association of this component to traffic.

### 3.3.3 Biomass burning


The third component (40 % of total variance) contained 82 % of the biomass burning markers (GAL, MANNO, LEV) and other combustion indicators, such as 46 % of the PAHs and 34 % of the mPAHs, including 79 % of the pinewood burning marker RET (Figure 3). Biomass burning is the main source for toxic PAHs, such as benzo[a]pyrene in the two background sites. It also substantially consists of oxy-PAHs, 9FLO, ANQ, BAF (24 % - 40 %) and 9nANT (32 %) which could be directly

emitted from biomass combustion, or other sources, or formed after oxidation of parent PAHs during atmospheric transport. However, there are very low loadings of other SOA markers, such as dicarboxylic acids and phthalic acid, suggesting that this component consists mainly of fresh biomass burning organic aerosols (BBOA) (Chuesaard et al., 2014; Medeiros & Simoneit, 2008).

Previous studies conducted in the urban area of Barcelona have always attributed BBOA to regional atmospheric transport to

city of Barcelona by land breezes through the surrounding river valleys during nighttime, essentially in winter (Alier et al., 2013; Brines et al., 2016; Reche et al., 2012; van Drooge et al., 2018). In agreement with these previous studies, biomass burning in the present study was almost exclusively related to the cold period (Figure 4), aligning with dominant contributions from domestic heating, and regional contributions from agricultural residue burning in fields, allowed in this Mediterranean region from mid-October to mid-March. Figure 4 shows that the score values for BBOA in the cold period were similar among

the samples in the two background sites, although higher scores were obtained in the city site during the nighttime, evidencing





that this introduction of biomass burning aerosols into the urban airshed, possibly by the land breezes or by direct contributions from the outskirts of the city. Local nighttime contributions may also influence the elevated site, since the score values in this site do not decrease substantially with respect to daytime levels (Figure 4). At night, the two air masses were physically separated due to a low boundary layer, so the similar score values indicate fresh biomass burning emissions. A separate multi-

variant analysis of the two datasets of the sites did not change the composition or intensity of the loadings and scores, neither could any further component be resolved dividing fresh from aged biomass burning aerosols. This indicates that biomass burning is probably a very substantial emission source in urban and sub-urban backgrounds within the metropolitan area of Barcelona during the cold period, and the main source of toxic PAHs, such as benzo[a]pyrene (Figure 4).

### 3.3.4 Biogenic

The fourth component (14 %) was mainly composed by biogenic organic aerosol markers, such as those from organic soil dust and detritus (alpha and beta glucoses, 96 and 97 %) and fungal spores (mannitol, sorbitol and meso-erythritol, 59 %, 42 % and 40 %, respectively) as well as high loadings from the isoprene oxidation products (49 %), malic acid (50 %) and succinic acid (31 %; Figure 3). Therefore, it is representing biogenic POA, with influence of SOA. Beside the presence of methyltetrols, short-chained dicarboxylic acids may be due to the oxidation of isoprene, which has been observed in laboratory and field

studies (Altieri et al., 2008; Bikkina et al., 2021) through aqueous-phase reactions (Ervens et al., 2011).

The scores of this component have a clear seasonality with high score values in the warm period (Figure 4). Moreover, in this period, scores are similar in the two sites except in the daytime sample from May 3rd in the elevated site, which may be due to the short rain episode on this day (Figure S4). High relative humidity conditions are associated with higher levels of pollen (Ila Gosselin et al., 2016; Rathnayake et al., 2017), which may be captured in this sample.

### 3.3.5 Regional secondary

The fifth component (29 % of explained variance) is dominated by secondary organic compounds, such as dicarboxylic acids (39 %) and biogenic SOA compounds (2MGA, 2MT1, 2MT2, 3HGA and MBTCA; 37%) (Figure 3). This component is related to aged SOA that recirculates in the regional atmosphere, with substantial chemical loadings of PM$_{10}$ (16 %) and ozone (29 %). This component is dominant in the samples from the warm period, consistently with higher biogenic VOC emission, higher

temperatures and solar radiation (Figure S4).

The higher score values were observed on days with continental air masses (Figure S3 b) supporting a regional origin of this aged secondary component. In general, the score values in the two sites followed similar trends (Figure 4). However, during the day, mixing was more uniform in the urban airshed, while at night, scores were higher at the elevated site, indicating either faster depletion in the city, or long-range atmospheric transport and formation within the ozone reservoir above the urban

airshed (Jaén, Udina, et al., 2021).





### 3.3.6 Fresh secondary

The sixth component (16 %) had high loadings of cis-pinonic acid (84 %) and $O_3$ (52 %) (Figure 3). This component was generally more abundant at the elevated site, where the sampling station has pine forest nearby and, therefore, α-pinene and its oxidation derivative, CPA (Figure 4). The higher abundance of this component in the winter campaign may be explained by lower humidity, which may preserve CPA over long time periods before transformation into further oxidation products, as has been observed in previous studies in the city (van Drooge et al., 2018).

This component was more abundant in the cold period campaign following an opposite trend than the above-described Regional secondary component that dominated in the warm period. In the elevated site, the Fresh secondary component was predominant in the daytime samples, which is in agreement with favoured photochemical reactions and nucleation processing (Alier et al., 2013; Dall'Osto et al., 2013). However, the high scores also observed at night indicate a conservation of this aerosol and ozone in the upper layers.

### 3.4 Source contributions and seasonal variability

As expected, NO and $NO_2$ were highly represented in the traffic components, comprising 83 % and 68 % of the total loadings of these compounds, respectively. $O_3$ was related with secondary components, 52 % and 29 % for the fresh and regional, respectively. $PM_{10}$ was represented in multiple components, namely secondary (45 %) but also traffic (22 %) and Biomass Burning (17 %).

The highest PAH loading (46 %) was related to biomass burning component which contrasts with observations one decade ago that attributed these pollutants to traffic emissions (Alier et al., 2013; van Drooge et al., 2018). Although these previous studies also included urban traffic sites, a general decrease of benzo[a]pyrene in the Air Quality network has been detected in urban traffic sites (-73 % between period 2010-2015 and period 2020-2023), while there is a smaller decrease in urban background sites (-28 %). This change reflects improvement in engine efficiency of motorized vehicles and the traffic restrictions in the low emission zone, in combination with an increase of biomass burning emissions during the past decade in the background sites. In contrast, the sum of the two traffic components represents the highest loadings of the PAHs derivatives, they account for the 56 % of mPAH (excluding RET), the 46 % of oxyPAHs and the 62 % of nitro-PAHs. In fact, the individual oxidized compounds show some significant differences in their abundances among the components. The more volatile oxyPAHs, 9FLO and ANQ, are probably mainly secondary, formed with the Fresh secondary aerosol formation while 23dmANQ and BAF are associated with the diurnal traffic. For the nitro-PAHs, 2nFLU is probably secondary formed after diurnal traffic emissions (46 %) while 9nANT is predominantly related to nocturnal traffic (51 %).

The polar fraction, excluding saccharides, is related with the secondary and biogenic components. In particular, most of the DCA and oleic acid have a regional origin, 39 % and 43 %, respectively, while biogenic SOA compounds (isoprene and α-pinene SOA) are more uniformly distributed between the non-combustion sources. Finally, the glucoses and fungal markers have almost solely relevant loadings from the biogenic component with percentages from 42 to 97 % in this component.



The relative contribution from each component to the sum of scores varied significantly by season, site and period of the day (Figure S7). During the warm campaign, traffic-related components dominated at the city site, contributing 40 % during the day and 49 % at night, though regional contributions were also substantial (29 % day; 36 % night). In contrast, at the elevated site, traffic contributions were lower during the day (17 %) and nearly absent at night (4 %), with the Regional Secondary component becoming dominant (35 % day; 67 % night). The biogenic POA component was also relevant, particularly at the elevated site (32 % day; 13 % night) compared to the city site (18 % day; 10 % night). On the other hand, in the cold period, biomass burning become the dominant component in all cases, especially at night. At the city background site, it accounted for 56 % of the scores during the day and 72 % at night. A similar dominance was observed at the elevated site, where it accounted for 57 % of the scores during the day and 62 % at night, highlighting the influence of biomass burning in urban background areas with an implication of local sources. The Fresh secondary component also had relevant contributions to the sum of scores during the cold campaign, particularly at the elevated site (30 % day; 22 % night) although it was also important in the city site during the day (19 %).

### 3.5 Vertical distribution of OA

Significant correlations (p < 0.01) between score values in the samples from the urban background site at city level and the elevated site were observed for Biomass burning, Regional secondary and Biogenic components ($r^2$=0.86, 0.89 and 0.56, respectively; Figure S8) with similar score values at both sites which suggests vertical and horizontal mixing, or local primary emissions, such is the case of nighttime biomass burning emissions in the elevated site. The Diurnal traffic component was also significantly correlated between sites ($r^2$=0.71), but shows 3 times higher scores at the city site compared to the elevated site (Figure S8). The Fresh secondary also showed good correlation between sites ($r^2$=0.42) but with higher scores at the elevated site, especially when the scores were low in the city site (Figure S8). This behaviour indicates a common influence between the two sites for the distribution of these pollutants but pointing to specific sources and processes for their origin, e.g., Diurnal traffic in the city and Fresh secondary related to nucleation at the elevated site.

On the contrary, the Nocturnal traffic component does not show a significant correlation between sampling sites with substantial higher scores at the city site. This decoupled behaviour between altitudes suggests the accumulation of this traffic related component in the nocturnal urban atmosphere in absence of turbulent mixing in nighttime period. The score values of the biomass burning in the urban area at city level in the cold period were anti-correlated with the planetary boundary layer height (PBLH; $r^2$ = 0.63; p = 0.003; Figure 5). This anti-correlation was not observed at the elevated site, and suggest local emissions in the elevated site, and input of biomass burning emissions and accumulation in the urban airshed for the city site.

### 3.6 SOA formation mechanisms

The Fresh secondary component was significantly anti-correlated with relative humidity in all cases (Figure 5). These correlations evidenced the importance of low RH in the formation through nucleation and conservation of fresh SOA (Brines et al., 2019; van Drooge et al., 2022). The chemical compounds involved in the formation of this fresh SOA likely result from



reaction with O₃, which is also dominantly present in this component. On the other side, the formation of the secondary compounds present in the diurnal and nocturnal traffic components could be more related with the oxidation with the $NO_2$ as evidenced by the presence of nitro-PAHs in these components.

Contrarily to Fresh secondary, the Regional secondary component was significantly correlated with relative humidity in the city site in the warm period and elevated site in both periods (Figure 5). The formation of the chemical compounds contributing

to this SOA component involved aqueous-phase oxidation under high relative humidity. The formation of SOA compounds from biogenic VOCs is influenced by multiple factors, including atmospheric conditions, aerosol acidity, and the presence of oxidants such as $NO_2$ (Surratt et al., 2010). Unlike Fresh and Regional secondary components, individual SOA compounds from isoprene oxidation do not exhibit significant correlations with RH (Figure S9) which is characteristic of non-acidic conditions (Nestorowicz et al., 2018) and aligning with the little influence of RH in isoprene SOA yields (Dommen et al.,

2006). However, their relative composition (2MGA/∑2MT) shows an anticorrelation with ambient humidity with independence from site and sampling periods, suggesting that different isoprene SOA formation pathways dominate under varying RH conditions. Similarly, the correlation of isoprene SOA compounds with $NO_2$ is significant only for samples from the city site during the spring campaign, where concentrations decrease with increasing $NO_2$ (Figure S10). Notably, the 2MGA/∑2MT ratio exhibits opposite trends with $NO_2$ in the warm and cold campaigns. In the warm campaign, higher

2MGA/∑2MT ratios are observed with increasing $NO_2$, aligning with literature that describes the preferential formation of 2-methylglyceric acid under high NOx conditions (Surratt et al., 2010). In contrast, during the cold campaign, the ratio decreases as $NO_2$ levels rise.

For α-pinene SOA, the observed trends resemble those of the Fresh and Regional components, with increased aged-to-fresh ratios ((MBTCA+3HGA)/CPA) under higher RH conditions (Figure S9). In contrast, $NO_2$ shows a negative correlation with

this ratio only in the city site during the warm campaign (Figure S10).

These opposite trends of individual SOA compounds with RH and $NO_2$ may suggest a relatively minor contribution from local biogenic SOA formation and a greater influence from regional transport. In fact, notable differences in SOA concentrations and relative abundances are observed under different air mass regimes (Figure S11), which also reflect variations in RH. Isoprene SOA concentrations are higher in samples influenced by Mediterranean (Regional) and Continental air masses (dark

blue and light blue clusters (Figure S3 b)).

## 4 Conclusions

The molecular analysis of organic compounds in $PM_{10}$ at two altitudes in Barcelona allowed a description of the POA distribution, the discernment of different SOA formation pathways and an assessment of their impacts upon the air quality in the city complementing previous studies in the area. Of all 68 analysed organic molecular markers, the biomass burning

markers, hopanes and PAHs, and most of their derivatives were significantly higher in the urban background site at city level



(81 m asl), the dicarboxylic acids were not clearly related with a specific site while the biogenic compounds were more abundant at the elevated site (415 m asl).

The bilinear decomposition of the combined dataset (organic compounds with NO, $NO_2$, $O_3$ and $PM_{10}$) with the MCR-ALS algorithm resolved 6 components that explained 95 % of the variance of the dataset. Those components were; 1) Diurnal traffic, with high loadings of the traffic markers (hopanes), NOx, and a contribution from harbour emissions, 2) Nocturnal traffic, related with gasoline and diesel combustion emissions, 3) Biomass Burning, containing levoglucosan and PAHs, 4) Biogenic, with primary and secondary biogenic compounds, 5) Regional secondary, with high contributions of aged SOA and 6) Fresh secondary mainly composed of cis-pinonic acid that could be related to new particle formation.

The main contribution to the parent PAHs was from biomass burning while many PAH derivatives were associated with the traffic components. However, some particular cases are of especial relevance as 9-fluoreneone and 9,10-anthraquinone appeared to be mainly secondary, formed with the Fresh secondary component, and 2-nitrofluoranthene which was part of the SOA formed from diurnal traffic emissions in the urban airshed. On the other hand, despite in urban areas usually being more associated with primary emissions, the dicarboxylic acids present in Barcelona were essentially due to recirculation of the regional air masses. As a consequence, SOA was more abundant at the elevated site while aerosols from traffic emissions were more abundant at the lower city level. Nocturnal traffic aerosols were more abundant in nighttime samples at the city site indicating an accumulation of these emissions in the urban airshed.

All component's score values showed good correlations between sites except for Nocturnal traffic evidencing disconnected air masses at both altitudes during nighttime periods, confirmed by different wind directions. Biomass burning was detected in higher abundance during the cold period, and at similar levels the two background sites, indicating local emissions in the elevated site as well as an accumulation of biomass burning aerosols in the city site during nighttime.

The relationship with atmospheric variables revealed different SOA formation mechanisms. The Fresh secondary component was temperature dependent in all sites and was enhanced under drier conditions, pointing to nucleation processing. On the contrary, the Regional secondary component, including isoprene and aged α-pinene showed positive correlations with relative humidity suggesting an aquatic phase oxidation.

**Author contributions**

C.J.: Sampling, Formal Analysis, Methodology, Investigation, Visualization, Writing – Original Draft Preparation, Writing – Review & Editing; M. U.: Methodology, Writing – Review & Editing; R. H: Methodology, Investigation, Writing – Review & Editing; J.O.G.: Conceptualization, Funding Acquisition, Project Administration, Supervision, Writing – Review & Editing; B.L.D: Sampling, Formal Analysis, Methodology, Investigation, Writing – Review & Editing;;



**Acknowledgments**

Acknowledgments to Roser Chaler and Alexandre Garcia for GC-MS technical assistance, and to Alfons Puertas for technical assistance at the Fabra Observatory. This study was funded by the Spanish Ministry of Science and Innovation's INTEMPOL project (PGC2018-102288-B-I00), DINAMIQS project (PID2022-140392OB-I00) and European Commission project PARC
(HORIZON-HLTH-2021-ENVHLTH-03: 101057014). C.J. thanks the FPU 19/06826 grant.

**Competing interests**

The authors declare that they have no conflict of interest.








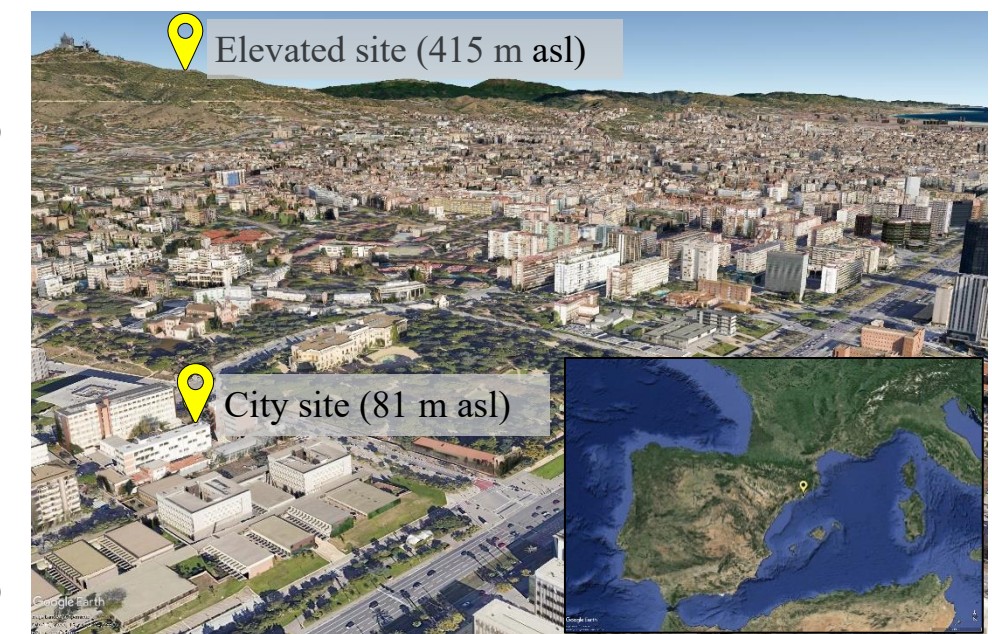

Figure 1: Sampling sites in Barcelona. Source: © Google Earth Pro.



**Figure 2: Average decreasing indexes (DI) for day and night sampling periods. Negative DI indicate higher concentration at the city site. Abbreviations are defined in Table 1.**





**Figure 3: MCR-ALS compound loadings (bars) for all components of the multivariate analysis. Purple dots show the percentage of the compound in a specific component in relation to all components. Abbreviations of compounds are defined in Table 1.**






**Figure 4: MCR-ALS component scores for the two sampling periods at both sampling sites. Dashed lines indicate the start and end of weekends. Yellow and grey shadowed areas indicate day and night samplings, respectively.**



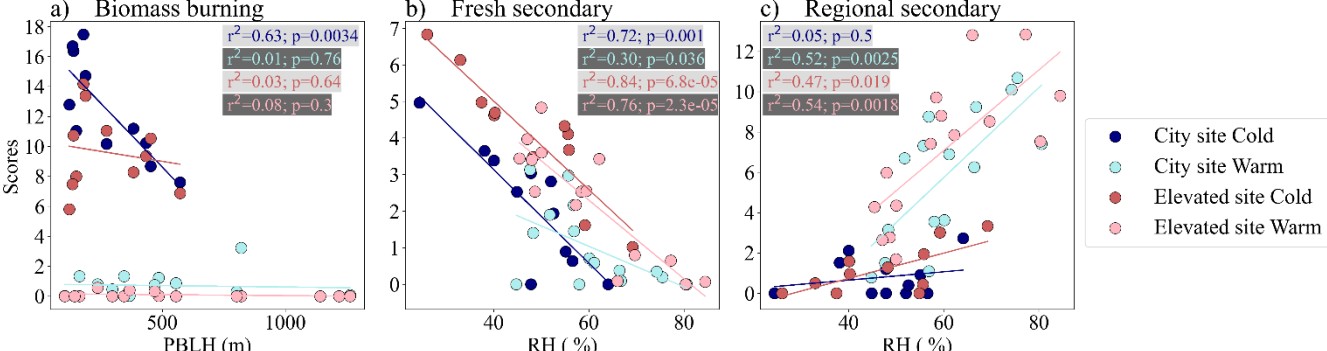

**Figure 5. Correlation of some component scores with different variables. PBLH: Planetary boundary layer height. RH: Relative humidity.**





**Table 1. Table containing compounds, abbreviations, detection frequencies (DF), method limit of detection (MLOD), average, maximum and minimum concentrations of the studied compounds at both sites and p-values for the Wilcoxon test for Paired Samples (WPS) between sites.**

| | | | DF | MLOD | City (81 m asl, urban) | | | Elevated site (415 m asl, sub-urban) | | | WPS |
|---|---|---|---|---|---|---|---|---|---|---|---|
| | | | | | Avg | Max | Min | Avg | Max | Min | p-val |
| | | | % | | pg/m³ | | | | | | - |
| PAH (n=20) | Phenanthrene | PHE | 89 | 12 | 58 | 120 | 14 | 42 | 113 | BLD | <0.05 |
| | Anthracene | ANT | 80 | 1 | 8 | 18 | BLD | 6 | 18 | BLD | <0.05 |
| | Fluoranthene | FL | 100 | 3 | 135 | 587 | 28 | 84 | 224 | 10 | <0.05 |
| | Pyrene | PYR | 100 | 4 | 147 | 334 | 28 | 97 | 255 | 12 | <0.05 |
| | Benzo[a]anthracene | BAA | 100 | 0.4 | 53 | 125 | 5 | 34 | 103 | 2 | <0.05 |
| | Triphenylene+Chrysene | C+T | 100 | 0.5 | 132 | 334 | 17 | 87 | 247 | 8 | <0.05 |
| | Benzo[b + j + k]fluoranthene | BBJKFL | 100 | 4 | 230 | 569 | 27 | 152 | 433 | 11 | <0.05 |
| | Benzo[e]pyrene | BEP | 100 | 2 | 125 | 305 | 20 | 80 | 232 | 6 | <0.05 |
| | Benzo[a]pyrene | BAP | 100 | 2 | 94 | 260 | 8 | 67 | 217 | 4 | <0.05 |
| | Perylene | PYL | 100 | 0.5 | 18 | 57 | 1 | 12 | 51 | 1 | <0.05 |
| | Dibenz[a,j]anthracene | DBAJANT | 94 | 0.7 | 30 | 94 | BLD | 22 | 65 | BLD | <0.05 |
| | Indeno[123cd]pyrene | IP | 100 | 2 | 127 | 300 | 25 | 82 | 222 | 11 | <0.05 |
| | Dibenz[a,h+a,c]anthracene | DBAH+ACANT | 100 | 0.7 | 19 | 49 | 1 | 13 | 35 | 1 | <0.05 |
| | Benzo[ghi]perylene | BGHIP | 100 | 2 | 192 | 391 | 33 | 106 | 263 | 12 | <0.05 |
| | Anthanthrene | ANTHAN | 100 | 0.4 | 39 | 87 | 10 | 28 | 67 | 5 | <0.05 |
| | Coronene | COR | 76 | 12 | 57 | 120 | BLD | 29 | 69 | BLD | <0.05 |
| mPAH (n=13) | 3-Methylphenanthrene | 3mPHE | 100 | 1 | 15 | 70 | 4 | 9 | 17 | 2 | <0.05 |
| | 2-Methylphenanthrene | 2mPHE | 93 | 3 | 28 | 73 | 4 | 12 | 27 | BLD | <0.05 |
| | 2-Methylanthracene | 2mANT | 100 | 0.2 | 5 | 20 | 1 | 2 | 9 | 0.2 | <0.05 |
| | 9-Methylphenanthrene | 9mPHE | 89 | 2 | 12 | 60 | 4 | 6 | 13 | BLD | <0.05 |
| | 1-Methylphenanthrene | 1mPHE | 100 | 0.8 | 14 | 43 | 3 | 10 | 31 | 1 | <0.05 |
| | 1,7-Dimethylphenanthrene | 17dmPHE | 100 | 0.4 | 16 | 37 | 3 | 12 | 33 | 1 | <0.05 |
| | 1+3-Methylfluoranthene | 1+3mFL | 100 | 0.1 | 34 | 89 | 3 | 21 | 61 | 2 | <0.05 |
| | Retene | RET | 100 | 3 | 183 | 568 | 9 | 157 | 487 | 5 | <0.05 |
| | 4-Methylpyrene | 4mPYR | 100 | 0.1 | 25 | 139 | 4 | 12 | 29 | 1 | <0.05 |
| | 1-Methylpyrene | 1mPYR | 100 | 0.1 | 18 | 76 | 2 | 9 | 23 | 1 | <0.05 |
| | 3-Methylchrysene | 3mC | 100 | 0.1 | 53 | 126 | 13 | 27 | 65 | 5 | <0.05 |
| | 6-Methylchrysene | 6mC | 100 | 0.1 | 10 | 22 | 2 | 5 | 11 | 1 | <0.05 |
| oxyPAH (n=5) | 9H-Fluoren-9-one | 9FLO | 65 | 22 | 67 | 155 | BLD | 61 | 173 | BLD | 0.18 |
| | 9,10-Anthraquinone | ANQ | 100 | 9 | 314 | 745 | 92 | 289 | 660 | 60 | 0.07 |
| | 2-Methylanthraquinone | 2mANQ | 100 | 1 | 128 | 240 | 79 | 82 | 217 | 42 | <0.05 |
| | 2,3-Dimethylanthraquinone | 23dmANQ | 100 | 2 | 19 | 120 | 7 | 10 | 52 | 3 | <0.05 |
| | Benzo[a]fluorenone | BAF | 100 | 1 | 63 | 296 | 18 | 37 | 88 | 10 | <0.05 |
| nitroPAH (n=2) | 9-Nitroanthracene | 9nANT | 94 | 0.6 | 11 | 33 | 2 | 5 | 18 | BLD | <0.05 |
| | 2-Nitrofluorene | 2nFLU | 61 | 0.8 | 2 | 8 | BLD | 2 | 6 | BLD | <0.05 |
| Hopanes (n=2) | 17a(H)21β(H)-29-norhopane | norHOP | 100 | 8 | 119 | 319 | 30 | 44 | 200 | 8 | <0.05 |
| | 17a(H)21β(H)-hopane | HOP | 100 | 5 | 115 | 320 | 28 | 40 | 207 | 7 | <0.05 |
| | | | | | ng/m³ | | | | | | - |
| Biomass burning tracers (n=3) | Galactosan | GAL | 100 | 0.02 | 6.3 | 23.9 | 1.1 | 5.5 | 21.2 | 0.5 | <0.05 |
| | Mannosan | MANNO | 100 | 0.1 | 5.8 | 21.4 | 1.1 | 4.7 | 18.3 | 0.4 | <0.05 |
| | Levoglucosan | LEV | 100 | 0.2 | 79.9 | 312.9 | 11.4 | 65.9 | 271.4 | 4 | <0.05 |
| Dicarboxylic acids (n=11) | Oxalic acid | OXA | 98 | 0.4 | 4.7 | 9.4 | 1.0 | 4.8 | 11.0 | BLD | 0.92 |
| | Malonic acid | MALA | 91 | 0.1 | 1.7 | 4.9 | BLD | 2.5 | 7.5 | BLD | <0.05 |
| | Succinic acid | SA | 100 | 0.8 | 13.5 | 24.7 | 7.2 | 13.3 | 33.9 | 6 | 0.67 |
| | Glutaric acid | GLU | 98 | 0.6 | 2.8 | 5.7 | 1.2 | 2.9 | 6.2 | 1 | 0.84 |





| | | | | | | | | | | |
|---|---|---|---|---|---|---|---|---|---|---|
| | Adipic acid | ADIA | 91 | 0.4 | 0.8 | 1.7 | BLD | 0.7 | 1.5 | BLD | 0.12 |
| | Pimelic acid | PIMA | 81 | 0.2 | 0.3 | 0.7 | BLD | 0.3 | 0.4 | BLD | <0.05 |
| | Suberic acid | SUBA | 81 | 0.4 | 0.7 | 1.7 | BLD | 0.5 | 0.9 | BLD | <0.05 |
| | Azelaic acid | AZA | 96 | 2 | 5.1 | 10.7 | 2.4 | 3.3 | 7.9 | 2 | <0.05 |
| | Malic acid | MA | 100 | 0.2 | 17.1 | 36.1 | 3.3 | 21.0 | 95.7 | 2 | 0.09 |
| | Tartaric acid | TARA | 100 | 0.01 | 3.5 | 8.3 | 0.2 | 4.3 | 11.4 | 0.1 | <0.05 |
| | Phthalic acid | PHA | 100 | 0.2 | 2.0 | 3.8 | 1.0 | 1.7 | 3.3 | 1 | <0.05 |
| Fatty acid (n=1) | Oleic acid | OLEA | 50 | 1 | 1.9 | 14.0 | BLD | 1.3 | 7.4 | BLD | 0.11 |
| Isoprene SOA (n=3) | 2-Metylglyceric acid | 2MGA | 100 | 0.2 | 3.0 | 6.5 | 1.0 | 3.5 | 7.0 | 1 | <0.05 |
| | 2-Methylthreitol | 2MT1 | 100 | 0.0 | 1.1 | 4.1 | 0.2 | 1.3 | 4.3 | 0.3 | <0.05 |
| | 2-Methylerythritol | 2MT2 | 100 | 0.1 | 3.2 | 12.9 | 0.7 | 3.7 | 14.8 | 1 | 0.69 |
| α-pinene SOA (n=3) | Cis pinonic acid | CPA | 100 | 0.2 | 14.3 | 59.3 | 3.1 | 27.0 | 134.2 | 3 | <0.05 |
| | 3-Hydroxyglutaric acid | 3HGA | 100 | 0.02 | 6.8 | 15.0 | 1.1 | 7.0 | 16.5 | 1 | 0.37 |
| | 3-Methyl-1,2,3-butanetricarboxylic acid | MBTCA | 98 | 0.1 | 4.5 | 10.5 | 1.0 | 4.8 | 12.6 | 1 | <0.05 |
| Other saccharides (n=5) | Mannitol | MANNI | 100 | 0.2 | 17.4 | 46.8 | 1.9 | 26.5 | 93.1 | 2 | <0.05 |
| | Sorbitol | SOR | 100 | 0.1 | 0.8 | 2.0 | 0.2 | 0.8 | 2.5 | 0.1 | 0.39 |
| | Meso erythritol | MERY | 100 | 0.2 | 6.0 | 12.7 | 2.5 | 5.6 | 12.2 | 1 | 0.24 |
| | Alpha-glucose | AGLU | 100 | 0.8 | 20.8 | 86.4 | 3.2 | 33.2 | 289.4 | 3 | <0.05 |
| | Beta-glucose | BGLU | 100 | 0.9 | 24.4 | 100.1 | 3.3 | 41.7 | 375.9 | 3 | <0.05 |



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
