# Peer review of "Origin, transport and processing of organic aerosols at different altitudes in coastal Mediterranean urban areas"

_EGUsphere, 2025_

## Author Comment (AC1)

**Reply to comments of reviewer 1:**

Comments to the manuscript titled "Origin, transport and processing of organic aerosols at different altitudes in coastal Mediterranean urban areas" by Clara Jaén et al.

The authors measured 68 organic molecular markers in diurnal and nocturnal $PM_{10}$ in an urban background site and in a nearby elevated sub-urban background site, in cold and warm periods in Barcelona. By applying MCR-ALS method, six different sources were resolved based on the profile of organic molecular markers. This manuscript has a good discussion of the influence of sources, altitude, seasons and diurnal differences. However, there are some concerns about the source apportionment methods and stability of the results.

1) Firstly, the manuscript lacks the basic information of the MCR-ALS method and the rational of choice among the other source apportionment methods, e.g., Positive Matrix Factorization (PMF).

2) Secondly, two weeks of diurnal and nocturnal sampling in cold and warm periods will have 28 samples, which is far behind the substance number (i.e., 67) input in the model. Therefore, the stability of results and the choice among different runs will be a big concern for the solidity of results of discussion. For example, for receptor models, depending on the number of degrees of freedom per variable, the suggested minimum number of samples (N) needs is $N > 30 + (V+3)/2$, where V represents the number of species and should be 65 here in the study (Henry et al., 1984). PMF has several sensitivity analysis, i.e., factor choosing, bootstrapping (BS) mapping and displacement (DISP) diagnostics. Please indicate the similar analysis for MCR-ALS to make sure the source apportionment result is solid and strong enough to support the subsequent discussions.

3) Thirdly, given the limited samples number involved in the analysis, some interpreted source is not trusty, e.g., diurnal and nocturnal traffic and fresh secondary. Specially, for fresh secondary, the sampling matric, i.e., $PM_{10}$ is far from the new particle generation (e.g., $PM_1$ or smaller).

4) Fourthly, the interpretation is based on source indicators like PAHs, hopanes, retene and others but in the real environment most of them are not single-source specific, fossil fuel combustion, vehicle emission and biomass burning will be integrated (Bi et al., 2008; Shen et al., 2012). Therefore I suggest the authors to reduce the number of sources and make the results more solid. Other multivariate analysis that don't have strict limit on degrees of freedom, i.e., unsupervised hierarchical clustering analysis, could be supplemented to support the source apportionment results. Also, the source apportionment of other studies conducted in Barcelona should be compared with and discussed in the

manuscript in detail. Given the above main concern, for now the manuscript is not qualified for ACP but it could be considered after major revision.

We thank the reviewer for pointing this out. The lack of information on the MCR-ALS method was due to a formatting error during submission, which caused this section of the methodology to be omitted without our awareness. We apologize for this error, and we have now restored this section (2.6) in the revised manuscript. In this part, we describe the fundamentals of the MCR-ALS approach (matrix decomposition under non-negativity constraints) and explain its advantages in comparison to other source apportionment methods, such as CMB, PCA, and PMF. This section also explains our choice of MCR-ALS, as it has been successfully applied in previous studies on atmospheric particulate matter and organic aerosols.

Regarding the sample-to-variable ratio, techniques such as PCA and PMF can be limited because their algorithms define factors based on covariance and may become unstable when the ratio falls below 3. In contrast, MCR-ALS is an iterative optimization method that does not rely on variance maximization. Its stability depends primarily on data quality, the constraints applied, and the number of components selected. Unlike PCA and PMF, MCR-ALS does not impose strict restrictions on the initial matrix and is more flexible with smaller sample sizes. In fact, the method benefits from including a large number of variables, as this enhances the robustness of the resolved profiles. For this reason, MCR-ALS has been widely applied to large datasets, such as full-scan mass spectrometry data, to extract patterns from unknown compounds.

To ensure robust results, we checked the influence of using different numbers of variables. In addition to the full dataset, we tested a reduced matrix containing 42 source tracers, instead of 67, in order to comply the proposes criteria ($N>30+(V+3)/2$ (52 samples). The components from both approaches were broadly consistent, with only slight changes in explained variance (see next figures). The largest deviations appeared in the traffic-related components, which became more mixed in the reduced dataset. This confirms that including all quantified compounds provides a more comprehensive and reliable representation of the data. For transparency, we have attached a document with figures from the reduced matrix decomposition for comparison.

We agree that many of the compounds considered may originate from multiple sources, such as PAHs from different combustion sources, and therefore cannot be treated as unambiguous single-source markers. Moreover, despite being PM10 filter samples that were used in the present study, this fraction also contains particles that are of smaller aerodynamic diameter, such as PM1. In this sense, chemical loadings of components, such as the "fresh secondary" components, due to the abundance of cis-pinonic acid can be classified as such, since cis-pinonic acid is a first-generation reaction product of alpha-pinene

oxidation. This is precisely why we applied MCR-ALS, as this method groups compounds with similar temporal profiles, allowing for a more robust identification of sources by integrating their contributions. Reducing the number of components, as suggested, would lead to overly mixed profiles that are harder to interpret, while our approach preserves the distinction between sources as much as possible. In fact, complementary PCA analysis decomposes the dataset into 8 components, supporting the complexity of the mixture. Those eight PCA components display profiles similar to the six components identified in the MCR-ALS decomposition, with two of the MCR-ALS components representing combinations of two PCA components. We can include this comparison in the Supplementary Material if required (see figures R3 and R4).

The main findings of previous source apportionment studies in Barcelona are summarized in the introduction and further discussed in relation to our components in Sections 3.3–3.6. In the revised manuscript, we have added additional comparisons throughout the text to strengthen the discussion. It is important to note, however, that previous studies at both altitudes did not include organic composition, which makes our work complementary to and not directly comparable with those earlier studies.

[Figure]

*Figure R1. Loadings with reduced initial matrix.*

[Figure]

*Figure R2. Scores with reduced initial matrix.*

[Figure]

*Figure R3. MRC Loadings (columns) compared with PCA components (lines).*

[Figure]

*Figure R4. MRC Scores (columns) compared with PCA components (lines) for all samples.*

Specific comments:

1. Lines 49: The source of hopanes is not single.

We agree with the reviewer that hopanes are not always associated with a single source and can compounds in lubricant oils, and emitted by vehicles, or they can be emitted by coal (anthracite) combustion, as indicated by the proposed references in the former comments. However, coal combustion is not practised in Barcelona and this region of Europe. In our study, hopanes are primarily attributed to traffic emissions, which is consistent with the urban context. This interpretation is further supported by the source apportionment results, where hopanes are strongly associated with NO and $NO_2$ in the traffic components, reinforcing their origin in vehicular emissions. We clarified this in the manuscript (Line 265).

"Hopanes are related to traffic emissions for their presence in lubricant oils, although they are also emitted through coal combustion (Bi et al., 2008; Schauer et al., 2007). In the present study, hopanes are traffic markers due to the absence of coal combustion in the studied region,…"

2. Lines 116-117: Please indicate the sunset and sunrise time of sampling period and explain why the daytime and nighttime sampling period are chosen and different in cold and warm seasons.

The sampling campaigns were conducted using UTC time, which resulted in a one-hour difference in local time between the cold and warm seasons. To maintain consistency, we applied the same UTC-based schedule, ensuring fixed 12-hour daytime and nighttime sampling periods across campaigns. Although this approach prioritized comparability and operational simplicity, sunrise and sunset times were not directly aligned between seasons; the corresponding values are provided in the table below.

We are aware that this timing choice may influence the results, particularly for traffic-related components. Since traffic rush hours typically occur at the beginning and end of the sampling periods, this overlap may have contributed to the more challenging separation of traffic sources in the apportionment results. Nonetheless, we considered the standardized 12-hour sampling scheme the most robust strategy to compare daytime and nighttime periods across seasons.

|  |  | Warm | Cold |
|---|---|---|---|
| Local time | Sunrise | 6:52 | 8:01 |
|  | Sunset | 20:45 | 18:09 |
|  | Meridian | 13:48 | 13:05 |
| UTC Time | Sunrise | 4:52 | 7:01 |
|  | Sunset | 18:45 | 17:09 |
|  | Meridian | 11:48 | 12:05 |
| Total sun hours |  | 13:53 | 10:08 |
| Local time | Day Sample | 9h-21h | 8h-20h |
|  | Night Sample | 21h-9h | 20h-8h |
| UTC | Day Sample | 7h-19h | 7h-19h |
|  | Night Sample | 19h-7h | 19h-7h |

3. Lines 121-123: Please indicate which type of quantification method was used and explain why full scan methods was employed other than MRM methods.

The quantification method is described in the last paragraph of the Chemical Analysis section, which we have clarified in the revised manuscript (Line 144).

"Sixty-eight compounds were identified and quantified based on their chromatographic retention times, mass spectra and calibration with analytical internal standards and 5-point calibration curves."

A full-scan approach was employed for polar compounds by GC-MS, and parent PAHs and methyl- and oxy-PAHs, and hopanes by GC-Orbitrap-MS, because it provides a comprehensive overview of the chemical composition of the samples, enabling both targeted quantification and future non-target explorations. In addition, the limits of quantification achieved with this method were sufficiently low for our study objectives.

4. Lines 133-134: Please explain why different length of column was used to analysis different compounds.

Different column lengths were used to achieve optimum separation of the compound groups, 60 m column for polar compounds, and 30 m column for PAHs. This approach allowed us to optimize resolution, peak shape, and retention time for each group, and has been successfully applied in numerous studies. It has been clarified in the revised manuscript (Line 132).

"The chromatographic separation was obtained with a HP-5MS 60 m capillary column for the polar compounds and with a HP-5MS 30 m column for the non-polar, in order to achieve optimum separation compounds of each group."

5. Lines 141-144: Although there is only high-abundance nitro-PAHs are analyzed in this study, using two different instruments with two different source, EI and CI is weird. Please provide more evidence that they could provide equivalent results.

Although is not included in the manuscript we compared the two analyses of nitro-PAHs with 14 samples analysed with both procedures. Those samples were collected for 24-hour periods from 12/11/2022 to 25/11/2022 with the same instrument and filters described in the methodology. The sampling site was the Birmingham Air Quality Supersite located in the University of Birmingham campus. The concentrations obtained for 9-nitroanthracene were equivalent with an acceptable error (Following figure) while concentrations of 2-nitrofluorene were very low in both cases ($<4$ pg/m$^3$). Unfortunately, comparison for other nitro-PAHs such as 1-nitropyrene could not be achieved due to the bad performance in the GC-MS/MS analysis in those filters and are not included in the study.

[Figure]

*Figure R5. Concentrations obtained with both methodologies for 9-nitroanthracene.*

6. Lines 147-149: Please indicate the difference between method limit of detection (MLOD) and instrument limit of detection (ILOD) and provided detailed data in the Supporting Information given there are several instruments and methods involved in this study.

The method limit of detection (MLOD) refers to the entire analytical procedure, including sampling, extraction, and analysis, whereas the instrument limit of detection (ILOD) refers only to the instrumental performance, defined at a signal-to-noise ratio of 3. For nitro-PAHs, the highest ILOD of both instruments value was applied.

The MLOD values, which were used as the reference for quantification, are presented in Table 1. In cases where blank signals were very low or absent, the ILOD was used instead to ensure reliable quantification of GC–MS peaks.

We have clarified in the manuscript that (Line 147):

"The method limit of detection (MLOD), which reflects the entire analytical procedure including sampling, extraction, and analysis, was determined as the average blank level plus three times the standard deviation of blanks (Table 1). In cases where blank values were below the instrumental limit of detection (ILOD), the ILOD was applied instead. ILOD refers specifically to the instrumental performance and was set based on a minimum signal-to-noise ratio of 3 calculated from the lowest point of the calibration curve. To process the data, concentrations below MLOD were replaced by half of this value."

7. Lines 157-160: There might be some bias because the site difference of different data.

Meteorological data from the sampling locations were used as complementary information to support the interpretation of the PM characterization. Although the meteorological station for the urban site was not located directly adjacent to the sampling point, it is the official monitoring station for the area and shares the same urban background classification. We therefore considered it representative for describing the meteorological conditions. Using validated data from the official station also helps to minimize uncertainty and ensures consistency, even if a small degree of spatial variability cannot be completely excluded. Importantly, these meteorological data were used only as complementary information and have no direct influence on the main study outputs. The paragraph was re-written to (Line 158):

"These stations were located near the elevated and city sampling sites. The elevated station was adjacent to the corresponding sampling site, while the city station was situated approximately 1.2 km away from the PM sampling site (Figure S1). This spatial separation may introduce slight differences between the recorded meteorological parameters and those directly affecting the sampled air masses. However, the station shares the same urban background classification as the sampling site and was considered representative of the meteorological conditions."

8. Lines 180: It is hard to understand the temporal evolution of potential temperature ($\theta$).

The description of the temporal evolution of potential temperature ($\theta$) has been revised in the manuscript to improve clarity and readability. We have added an introductory sentence highlighting the relevance of $\theta$ for understanding vertical mixing and atmospheric stability and reorganized the paragraph to better distinguish daytime and nighttime dynamics at the two sites (Line 200):

"The temporal evolution of potential temperature ($\theta$) provides insight into the vertical mixing and stability of the atmosphere at the sampling sites. During both campaigns, $\theta$ exhibited similar trends at both altitudes (Figure S4). During the

day, temperature and relative humidity (RH) remained comparable at the two sites, indicating that they were within the convective mixing layer of the urban airshed. Temperature peaks at the elevated site were slightly delayed relative to the city site, reflecting the progressive growth of the mixing layer in the hours following sunrise. At night, more pronounced variation in temperature and RH were observed between the two sites, suggesting that the elevated site become partially decoupled from the urban airshed. Generally, θ was higher at the elevated site at night, consistent with the development of stable atmospheric conditions and temperature inversions. This day-night dynamic was also reflected in variations of wind speed and direction, showing similar patterns during the day but becoming stronger and originating from a different direction at the elevated site during the night compared to the city site. (Figure S4)."

9. Lines 189: It is confusing about the nocturnal decoupled behaviour for ozone.

The description of ozone behaviour has been clarified in the revised manuscript. We reorganized the paragraph to clearly distinguish daytime and nighttime dynamics and emphasized the observed decoupling at night between the city and elevated sites (Line 210).

"Regarding the air quality parameters, ozone exhibited a clear day-night contrast between sites. During the day, $O_3$ concentrations were all well correlated between the city site and the elevated site ($r^2$=0.7) (Figures S4 & S5), reflecting a well-mixed urban atmosphere. At night, this correlation weakened substantially, indicating a decoupling of the elevated site from the urban airshed. $O_3$ concentrations remained consistently high at the elevated background site throughout both day and night (78±15 μg/m$^3$), while at the city site, $O_3$ exhibited a diurnal oscillation peaking during sunlight hours (47±19 μg/m$^3$). These observations are consistent with previous studies in the same sampling sites across different seasons (Dall'Osto et al., 2013) and suggest that differences in oxidant distribution along the vertical column may influence the organic composition at the studied altitudes."

10. Line 223-226: Retene dosen't always indicate a single source, especially in different regions.

We agree with the reviewer that retene is not always related to a single-source, and that is can be emitted during incomplete combustion of pine wood (pine) and coal (anthracite), as shown in the proposed reference of Bi et al. 2008, and Shen et al. 2012 (added to reference list). However, coal combustion is not practised in Barcelona and this region of Europe, so in the present study retene is related solemnly to wood combustion, and strongly correlated to another wood burning indicator, levoglucosan, as described in 3.3.3 (Line 245).

"Retene (RET), which is an indicator of pinewood combustion in the absence of coal combustion (Bi et al., 2008; Ramdahl, 1983; Shen et al., 2012), was the more abundant methyl-PAH, with similar concentrations as predominant unsubstituted PAHs (pyrene (PYR), benzofluoranthenes (BBJKFL) and benzo[ghi]perylene (BGHIP)), which suggests contribution of pinewood combustion in the sites, since there is no coal combustion in the region (Van Drooge and Grimalt, 2015)."

11. Lines 231-232: It is not clear to attribute the concentration difference to different primary and secondary. Please explain in more detail.

The manuscript has been revised to provide a more detailed explanation and to indicate that the observed concentration patterns are suggestive of secondary formation or primary emissions (Line 253).

"The fact that their concentrations are similar at both sites suggest that secondary formation processes contribute substantially to their presence. In contrast, the other oxyPAH are significantly higher in concentration at the city level, indicating stronger influence of local primary emissions."

12. Lines 298: Please provide more information and method details about the MCR-ALS. Also, the difference of typical source apportionment tools like PMF should be explained in the discussion.

The authors apologize for the omission of a section in the methodology regarding the multivariate analysis. This information has now been included in the revised manuscript and refences were added to reference list (Line 183).

"Multivariate Curve Resolution—Alternating Least Square (MCR-ALS 2.0) was applied to the dataset (67 variables and 52 samples) using MATLAB (Jaumot et al., 2005, 2015; Tauler, 1995). The MCR-ALS method is based on the $D = CST + E$ matrix equation to decompose the initial matrix D (normalized dataset) into a reduced number of components. The output gives: C (a matrix with sample scores for each component), ST (a matrix with compound loadings for each component; profiles) and E (a matrix with residual non-explained data) The decomposition was performed under non-negativity constraints, which provide physically interpretable results and are more realistic for environmental data. The number of components was selected based on the interpretability of their chemical profiles in terms of emission sources and atmospheric processes.

Compared with other source apportionment approaches, such as Chemical Mass Balance (CMB), Principal Component Analysis (PCA), or Positive Matrix Factorization (PMF), MCR-ALS offers several advantages. CMB requires predefined source profiles and cannot resolve unknown sources, while PCA enforces orthogonality, which limits environmental interpretability. Both MCR-

ALS and PMF apply non-negativity constraints and yield comparable results, but they differ in their optimization algorithms and normalization procedures (Tauler et al., 2009). Importantly, MCR-ALS does not impose orthogonality between components, allowing for overlapping explained variance that better reflects the reality of atmospheric sources, which are rarely independent. Previous studies have demonstrated the robustness of this method for source apportionment of air pollutants and organic aerosols (van Drooge et al., 2022; Jaén et al., 2021b, 2023)."

13. Lines 235-238: The evidence is not strong enough to support the nocturnal traffic source.

This component shows highest score values in the nocturnal samples from the city site, and the loadings are represented by PAHs, mPAHs, and include hopanes, and NOx, and was therefore classified as "nocturnal traffic" component, as argumented in detail in section 3.3.2.

14. Lines 488-489: The evidence is not strong enough to support the new particle formation, especially due to the sampling matrix of $PM_{10}$.

Despite being PM10 filter samples that were used in the present study, this fraction also contains particles that are of smaller aerodynamic diameter, such as PM1. In this sense, chemical loadings of components, such as the "fresh secondary" components, can be classified as such due to the dominant presence of cis-pinonic acid from oxidation of α-pinene as explained in Line 294, and in 3.3.6.

Technical corrections:

1. Some font of citations is larger than others, e.g., Lines 39-40, Lines 225-226, Lines 255-256, 277-278, 347-348. Corrected

2. Lines 114: There should be one blank between number and unit. Corrected

3. Lines 121-123: The number should be subscript. Please check in the whole manuscript. Corrected

4. The language in the manscript needs to be modified by tools such as AI, e.g., Lines 186-188. Several parts of the manuscript, including the section mentioned, have been revised to improve grammar and readability.

5. Lines 235: The x of $NO_x$ should be subscript. Please scrutinize in the manuscript. Corrected

6. Lines 405-406: Confusing symbol before the number here.

The text has been revised to clarify the notation, replacing the confusing minus symbol with a clear description of the percentage decrease (Line 434).

---

## Author Comment (AC2)

**Reply to comments of reviewer 2:**

The authors conducted PM10 sampling (both day and night) at two different sites in Barcelona and performed chemical analysis to report molecular markers during two distinct seasons, which is commendable. I have a few concerns listed below:

1. Only 28 samples to run the source-apportionment analysis?

The total number of samples used in the source-apportionment analysis was 52, encompassing both sampling periods and both sites. As addressed in our response to another reviewer, we ensured the robustness of the results by performing the decomposition using different numbers of variables, which confirmed the stability and consistency of the identified sources. We clarified the number of samples in Line 111 and Line 183.

2. Description and logic behind using this apportionment method

We added an omitted section in the revised manuscript (2.6). In this section, we describe the fundamentals of the MCR-ALS approach (matrix decomposition under non-negativity constraints) and explain its advantages in comparison to other source apportionment methods, such as CMB, PCA, and PMF. References were added to the list. This section also explains our choice of MCR-ALS, as it has been successfully applied in previous studies on atmospheric particulate matter and organic aerosols (Line 182).

"Multivariate Curve Resolution—Alternating Least Square (MCR-ALS 2.0) was applied to the dataset (67 variables and 52 samples) using MATLAB (Jaumot et al., 2005, 2015; Tauler, 1995). The MCR-ALS method is based on the $D = CS^T + E$ matrix equation to decompose the initial matrix D (normalized dataset) into a reduced number of components. The output gives: C (a matrix with sample scores for each component), ST (a matrix with compound loadings for each component; profiles) and E (a matrix with residual non-explained data) The decomposition was performed under non-negativity constraints, which provide physically interpretable results and are more realistic for environmental data. The number of components was selected based on the interpretability of their chemical profiles in terms of emission sources and atmospheric processes.

Compared with other source apportionment approaches, such as Chemical Mass Balance (CMB), Principal Component Analysis (PCA), or Positive Matrix Factorization (PMF), MCR-ALS offers several advantages. CMB requires predefined source profiles and cannot resolve unknown sources, while PCA enforces orthogonality, which limits environmental interpretability. Both MCR-ALS and PMF apply non-negativity constraints and yield comparable results, but they differ in their optimization algorithms and normalization procedures

(Tauler et al., 2009). Importantly, MCR-ALS does not impose orthogonality between components, allowing for overlapping explained variance that better reflects the reality of atmospheric sources, which are rarely independent. Previous studies have demonstrated the robustness of this method for source apportionment of air pollutants and organic aerosols (van Drooge et al., 2022; Jaén et al., 2021b, 2023)."

3. L105:… PM10 was collected at two background sampling stations were installed at two altitudes .. needs rephrasing

The sentence has been rephrased for clarity and grammatical accuracy in the revised manuscript (Line 103).

"In the present study, PM10 samples were collected at two background stations at different altitudes (Figure 1) to assess how interactions between specific meteorological conditions and multisource particle emissions influence the organic composition of airborne aerosols in the city. One station, located at 81 m asl, represents the urban background site at IDAEA (city site), while the other, at 415 m asl, is situated atop the Collserola hills overlooking the city (elevated site). The two sites are separated by a horizontal distance of 3.5 km."

4. L116: Any justification for different times chosen for sampling?

As mentioned in our response to another reviewer, the sampling campaigns were conducted using Coordinated Universal Time (UTC), which resulted in a one-hour difference in local time between the cold and warm seasons. To maintain consistency, we adopted a fixed UTC-based schedule, ensuring standardized 12-hour daytime and nighttime sampling periods across all campaigns.

Although this approach prioritized comparability and operational consistency, sunrise and sunset times were not perfectly aligned between seasons. We are aware that this timing difference may have influenced certain results, particularly for traffic-related components, since rush hours often coincide with the beginning and end of the sampling periods. Nonetheless, we consider this standardized 12-hour scheme the most robust and reproducible strategy for comparing daytime and nighttime samples across different seasons.

5. L157: 'This information' means 'Meteorological parameters'? Moreover, it can be understood that the met station was not the same as the air quality monitoring station. Please mention this limitation in the text.

Indeed, "this information" refers to the meteorological parameters. To improve clarity, the text has been revised. Regarding the distance between sampling and meteorological station locations the limitation has been explicitly mentioned in

the revised manuscript. However, it is important to note that these meteorological data were used only as complementary information and have no direct influence on the main study outputs. The paragraph was re-written to (Line 158):

"These stations were located near the elevated and city sampling sites. The elevated station was adjacent to the corresponding sampling site, while the city station was situated approximately 1.2 km away from the PM sampling site (Figure S1). This spatial separation may introduce slight differences between the recorded meteorological parameters and those directly affecting the sampled air masses. However, the station shares the same urban background classification as the sampling site and was considered representative of the meteorological conditions."

6. L218: missing space after unit.

It has been corrected in the text.

7. L294: provide the reason for such results.

A possible reason for the reduction of nitro-PAHs and an increase of oxy-PAH may be attributable to reductions in $NO_2$ emissions in the city of Barcelona over the past decade. We added this to the text (Line 314).

"Regarding nitro-PAHs, few studies report their PM concentrations in southern Europe. In fact, the only measurements in ambient air in Barcelona were those by Bayona et al. (1994), conducted at heavily trafficked sites. They reported considerably higher concentrations of 9-nitroanthracene and 2-nitrofluorene than those observed in the present study, a pattern also reflected in the parent PAHs. In contrast, the oxy-PAH ANQ, which was also measured in that study, was found at lower concentrations. Studies conducted in European cities during the early 2010s similarly reported higher nitro-PAH concentrations than those observed here, particularly in winter (Alam et al., 2015; Alves et al., 2017; Tomaz et al., 2016), whereas more recent studies on the Iberian Peninsula reported comparable levels (Lara et al., 2022). Although further measurements are necessary, reduction of nitro-PAHs concentrations may be attributable to reductions in $NO_2$ emissions over the past decade, resulting from the implementation of low-emission zones and cleaner vehicle engines, which have decreased atmospheric formation of nitro-PAHs while potentially enhancing oxy-PAH formation through reactions with $O_3$."

8. Section 3.3: Some information must also be provided on 'already well-established' apportionment methods such as PMF.

In Section 2.6, we provide a comparison with other source-apportionment methods, including CMB, PCA, and PMF. Additionally, in section 3.3 we included discussions of our main findings in the context of previous source-apportionment studies in Barcelona that applied PMF. This allows readers to better understand the consistency of our results with established methods.

9. L325-6: 'methyl phenanthrenes' or 'methylphenanthrenes'?

The term has been unified throughout the manuscript to 'methylphenanthrenes' to align with the literature.

10. L430: 'Vertical' should be rephrased.

In order to avoid confusion, "Vertical" was substituted by "Altitudinal".